# Automation of Infectious Focus Assay for Determination of Filovirus Titers and Direct Comparison to Plaque and TCID_50_ Assays

**DOI:** 10.3390/microorganisms9010156

**Published:** 2021-01-12

**Authors:** Patrick T. Keiser, Manu Anantpadma, Hilary Staples, Ricardo Carrion, Robert A. Davey

**Affiliations:** 1Department of Microbiology, National Emerging Infectious Disease Laboratories, Boston University, Boston, MA 02215, USA; pkeiser@bu.edu (P.T.K.); manu.anantpadma@wuxiapptec.com (M.A.); 2Disease Intervention & Prevention, Texas Biomedical Research Institute, San Antonio, TX 78227, USA; hstaples@txbiomed.org (H.S.); rcarrion@txbiomed.org (R.C.)

**Keywords:** plaque assay, TCID_50_, FFU, filovirus, Ebola, Marburg virus, Sudan virus

## Abstract

Ongoing efforts to develop effective therapies against filoviruses rely, to different extents, on quantifying the amount of viable virus in samples by plaque, TCID_50_, and focus assays. Unfortunately, these techniques have inherent variance, and laboratory-specific preferences make direct comparison of data difficult. Additionally, human errors such as operator errors and subjective bias can further compound the differences in outcomes. To overcome these biases, we developed a computer-based automated image-processing method for a focus assay based on the open-source CellProfiler software platform, which enables high-throughput screening of many treatment samples at one time. We compared virus titers calculated using this platform to plaque and TCID_50_ assays using common stocks of virus for 3 major Filovirus species, *Zaire ebolavirus*, *Sudan ebolavirus*, and *Marburg marburgvirus* with each assay performed by multiple operators on multiple days. We show that plaque assays give comparable findings that differ by less than 3-fold. Focus-forming unit (FFU) and TCID_50_ assays differ by 10-fold or less from the plaque assays due a higher (FFU) and lower (TCID_50_) sensitivity. However, reproducibility and accuracy of each assay differs significantly with Neutral Red Agarose Overlay plaque assays and TCID_50_ with the lowest reproducibility due to subjective analysis and operator error. Both crystal violet methylcellulose overlay plaque assay and focus assays perform best for accuracy and the focus assay performs best for speed and throughput.

## 1. Introduction

Filoviruses are highly contagious and include *Ebolaviruses* and *Marburgviruses*. Both genera cause severe disease with *Ebolaviruses* causing outbreaks in western Africa from 2014 to 2016, as well as a more recent outbreak in Democratic Republic of Congo in 2019. Potential treatments and preventative measures are being tested, but is made difficult by the need to work in a Biosafety Level-4 (BSL-4) laboratory. Therefore, the need to use methods that can increase the throughput of research and reduce assay turnaround time during an outbreak is of importance.

Various detection and quantification methods for these viruses have been developed, including reverse-transcription polymerase chain reaction (RT-PCR), deep sequencing, transmission electron microscopy (TEM), 50% tissue culture infectious dose assay (TCID_50_), plaque assays, and viral antigen detection-based assays such as ELISA and immunostaining. During outbreak scenarios, methods such as RT-PCR or ELISAs are increasingly favored for diagnostics and detection. Although they are quick and sensitive, with RT-PCR detecting virus nucleic acids and ELISA, virus protein, they do not, however, determine the amount of infectious virus present in a sample.

Plaque assays, TCID_50_, and the focus-forming unit assay (FFU) are the most commonly used quantitative assays for titering infectious virus. Plaque and TCID_50_ assays both use the principal that the virus replication cycle produces cytopathology in cell monolayers to calculate virus titer. In a plaque assay, a highly viscous semi-solid or solid medium restricts virus diffusion from infected cells. The local replication of virus results in a change in cell viability and while for some viruses this can result in zones of clearing termed plaques, filoviruses most often cause a difference in staining by a dye [1,2]. The TCID_50_ assay differs by not restricting virus diffusion and results in gross cell death within a well of a tissue culture plate and detection of residual cells by a general cell stain [1]. For filoviruses, both the plaque and the TCID_50_ assays can take one to two weeks to complete. Furthermore, filovirus plaques tend to be small and irregular requiring a trained technician and introduces subjectivity in identifying and counting, causing increased variability in counts. In contrast, the FFU assay determines virus titer by fluorescent staining of the viral antigen expressed in cells during infection. As such, the FFU assay does not require cell death to measure virus titer, and takes only 2 days to complete. Additionally, with high content imaging becoming more common and high contrast of fluorescently stained cells, FFU assay counting can be automated, thereby removing operator subjectivity. 

Here we report the development of an FFU assay for isolates from 3 major filoviruses for quantitation of viable virus in a sample. Virus is quantified using automated image processing through the open-source software CellProfiler [3,4]. We compare outcomes using this assay to those from different plaque assays; Neutral Red Agarose Overlay Plaque Assay/FANG assay (NRAO) and Crystal Violet Methylcellulose Overlay Plaque Assay (CVMO) and the TCID_50_ assay. We compare performance, reproducibility, sensitivity, and technical variance between each assay.

## 2. Materials and Methods 

### 2.1. Virus and Cell Preparation

Vero E6 cells (ATCC) were maintained in Dulbecco Modified Eagle Medium (DMEM; Gibco, 11995073, Gaithersburg, MD, USA) supplemented with 10% Heat Inactivated Fetal Bovine Serum (Gibco, 10500064) (DMEM-10) at in a humidified incubator at 37 °C and 5% CO_2_ overnight. For Neutral Red Agarose Overlay (NRAO) plaque assays, cells were plated on 6 well tissue culture plates and taken into the BSL-4 at 90–100% confluency. For crystal violet plaque assays, cells were protected with 1% Penicillin/Streptomycin L-Glutamine (Lonza, 17-718R, Basel, Switzerland) in their DMEM-10 medium, plated on 6 well tissue culture plates, and taken into the BSL-4 at 75–90% confluency. TCID_50_ (Median Tissue Culture Infectious Dose) assays were plated on 96 well tissue culture plates in DMEM supplemented with 2% FBS (DMEM-2) and 1% Penicillin/Streptomycin L-Glutamine (Lonza, 17-718R) and allowed to grow overnight before entry into the BSL-4. After entry into the BSL4, plates were inoculated with either Ebola virus Zaire Mayinga (EBOV), MARV Marburg virus Musoke (MARV), or Sudan virus Gulu (SUDV). Focus-Forming Assay cells were plated on either 96 (180 µL in the first row, 100 µL in the remainder) or 384 well (25 µL in each well) plates with DMEM-10 and taken into the BSL4 at 75–90% confluency after overnight incubation. Plates were inoculated with either EBOV, MARV, SUDV.

### 2.2. Overlay Preparation

50 mL of NRAO primary overlay consisted of 25 mL Minimal Essential Medium Eagle with Earle’s BSS (EMEM, Lonza, 12-668E) buffered with 4% FBS, 2% L-Glutamine (Lonza, 17-905C), and 2% Sodium Pyruvate (Lonza, 13-115E) with 25 mL Agarose (1:1 mixture). 50 mL NRAO secondary overlay consisted of 25 mL EMEM buffered with 4% FBS, 2% L-Glutamine, 2% Sodium Pyruvate, and 8% Neutral Red (Gibco, special formulation) with 25 mL Agarose (1:1 mixture). Methylcellulose overlay consisted of 500 mL DMEM-2 with 1% Penicillin/Streptomycin L-Glutamine and 100 mL Methylcellulose (16.67%) (Sigma, M0387, St. Louis, MO, USA) All overlays were heated to at least 37 °C before adding. 

### 2.3. NRAO Assay Infection with Filovirus

Once inside the BSL-4, virus was retrieved, and a virus dilution series was prepared. 100 µL of virus was added to 900 µL DMEM-2, creating a ten-fold dilution. Virus was then serially diluted from 10^−1^ to 10^−6^ by transferring 100 µL virus/DMEM-2 across 6 tubes. Samples were mixed well using a new pipette between each dilution step. The DMEM-10 in the 6 well plates was then decanted into 5% Microchem solution. 400 µL of each dilution was plated in each well, beginning with the 10^−2^ to the 10^−6^ dilution. Adding 400 µL per well allowed for each dilution to be plated in duplicate. In the final well, 400 µL of DMEM-2 was added as an uninfected control. After virus was administered, plates were placed on a rocking platform within a humidified incubator at 37 °C, 5% CO_2_ for one hour.

After 1-h incubation, plates were removed from the incubator and inoculum was removed from each well by pipette and disposed of in 5% Microchem. Primary overlay was added at 2 mL in a drop-wise fashion to each well. The agarose overlay was allowed to solidify before incubating for 7 days at 37 °C, 5% CO_2_. After 7 days of incubation, 2 mL secondary overlay was added in a drop-wise fashion. Plates were placed back into the incubator overnight. The next day, plates were scanned using a flatbed scanner and scan files were assessed for viral titer. 

Wells with 15–150 plaques were considered within acceptable limits. Viral titer was calculated based on the number of plaques multiplied by the dilution factor and the correction factor of 2.5 (for 1 mL), and is reported in pfu/mL.

### 2.4. CVMO Assay

Once inside the BSL-4, virus was retrieved, and a virus dilution series was prepared. 100 µL of virus was added to 900 µL PBS, creating a ten-fold dilution. Virus was then serially diluted further from 10^−1^ to 10^−6^ by transferring 100 µL virus/PBS down 6 tubes. Samples were mixed well using a pipette between each dilution step. The DMEM-10 in the 6 well plates was then decanted into 5% Microchem solution. 400 µL of each dilution was plated in each well, beginning with the 10^−2^ to the 10^−6^ dilution. Adding 400 µL per well allowed for each dilution to be plated in duplicate. In the final well, 400 µL of PBS was added as an uninfected control. After virus was added, plates were placed on a rocking platform within a humidified incubator at 37 °C, 5% CO_2_ for one hour. 

After 1-h incubation, plates were removed from the incubator and inoculum was removed from each well by pipette and disposed of in 5% Microchem. Methylcellulose overlay was added at 5 mL in a drop-wise fashion to each well. The assay was then then incubated for 10 days at 37 °C, 5% CO_2_. After 10 days of incubation, methylcellulose overlay was removed from each well, plates were submerged in 10% formalin for inactivation and incubated overnight at 4 °C. After inactivation, plates were removed from the BSL-4 and washed with 1x PBS. Plates were next stained with 2 mL CVMO for 10 min at room temperature before washing excess staining material from the plate. Plaques were counted using a light microscope, with 25–250 plaques considered acceptable limits. Viral titer was calculated based on the number of plaques multiplied by the dilution factor and the correction factor of 2.5 (for 1 mL), and is reported in pfu/mL.

### 2.5. TCID_50_ Assay

Once inside the BSL-4, media was removed from the wells by decanting into 5% Microchem and washed with 200 mL PBS. The PBS wash was discarded and 180 µL DMEM-2/antibiotics was added to each well, with 20 µL virus placed into all wells of the first column. A 10-fold serial dilution series was completed by moving 20 µL increments down each row of the plate. Plates were then incubated 10 days at 37 °C, 5% CO_2_.

After incubation, plates were decanted into 5% Microchem. 100 µL CVMO stain was added to each well, and allowed to incubate at room temperature for 10 min. Plates were next decanted and thoroughly washed with water to remove excess staining material. Once dried, plates were assessed for cytopathic effect (CPE) in each column. The final titer was calculated using the Reed–Muench Method [5]. 

### 2.6. FFU Assay 

Inside BSL-4 containment, 20 µL (96 well) or 25 µL (384 well) of virus was added to each well of the first column. A half serial dilution series was completed by 100 µL (96 well) or 25 µL (384 well) increments down each row of the plate. Plates were then incubated 36–48 h in a humidified incubator at 37 °C, 5% CO_2_. 

After incubation, plates were decanted into 5% Microchem. Plates of both well types were submerged in 10% formalin for inactivation and incubated at 4 °C overnight. Plates were then removed from the BSL-4 and stained with antibody in 3.5% BSA and Hoechst 33,342. The concentration for Ebola Zaire primary antibody was 1:1500 with a mouse mAb anti-GP (IBT, 0201-020, Rockville, MD, USA). For Sudan virus, a concentration of 1:1000 was used in a mouse mAb anti-GP (IBT, 0280-001). Marburg was done at 1:3000 with a pVLP derived rabbit antibody (IBT, 01-0005). Cell nuclei were stained with Hoechst 33,342 at a 1:10,000 concentration. Plates where next imaged and then analyzed for virus titer using CellProfiler.

### 2.7. Statistics Analysis

Coefficients of variance were determined using R by dividing mean values by standard deviation of titers generated from virus aliquot replicates, converting to percentages, and were compared between assayers/counters [6]. Dataset used for titer and coefficient of variance calculations are included as Appendix A. Graphical analysis and statistical significance by one-way ANOVA, Turkey’s Multiple Comparison Test was determined using GraphPad Prism version 8.0.0 for Windows, GraphPad Software, San Diego, CA, USA.

## 3. Results

### Application of Automated Image Processing for Calculating Virus Titer Using FFU Assay

A significant hurdle for using the FFU assay for Filovirus titer calculation has been difficulty in counting infected cell foci. Foci are typically small, consisting of clusters of 2–5 cells and need to be imaged by a microscope. In the past we manually counted foci from individual images and so, were subject to operator error, in addition to being cumbersome. Others have been successful in adapting proprietary software, typically packaged with high end and expensive microscope imaging equipment [7]. To make FFU counting available to different laboratories; we applied CellProfiler, an open-source software platform that works on PC, MacOS and Linux operating systems that has a long track record for use in image-based analysis of cells [3,4]. Unlike other open-source image analysis software such as Fiji/ImageJ [8], CellProfiler is designed to analyze multiple images using a single, uniform automated algorithm (pipeline) and compile the data into a spreadsheet or database, making it better suited for analysis of large sets of assay images for most laboratories.

First, cells are identified by detection of cell nuclei stained with Hoechst 33,342, a dye that stains and fluoresces after binding cell DNA to visualize the cell nucleus. Virus antigen is detected by a specific primary antibody and fluorescently labeled secondary antibody. Images are filtered to detect each fluorophore and processed by CellProfiler. Image processing involves first adjusting the virus antigen detection threshold to eliminate any background signal. Although this can be done semi-automatically, it requires fine tuning by the operator and is best achieved by using Fiji/ImageJ to measure signal in the dark spaces (noise) versus that corresponding to infected cells (signal). A signal to noise ratio of 2 is sufficient but a larger ratio gives improved accuracy. Optimization of the signal is a function of the quality of the antibody, the optics of the microscope, and the camera. We recommend using a black and white 14 bit or better camera to provide sufficient image grey shades so that variation in staining quality does not impact image processing. Typically, we achieved a signal to noise ratio of >2. By optimization of cell density and using multi-channel pipettors we were able to perform the FFU assay in 96 or 384-well formats with similar outcomes. The latter provides significant advantages in throughput that are not easily obtained using plaque assays or TCID_50_. To read out the FFU assay, a manual or automated epifluorescent microscope is used to capture images of cells in wells of the plates. Image magnification did not alter counting significantly, with 2.5× through to 10× lenses adequately capturing thousands of cells per field. We prefer to use a 4× lens, but a magnification must be chosen that captures as much of the well as possible while avoiding edge artifacts in the images. The images for this manuscript were acquired using a Biotech Cytation Imager or a Nikon Ti2 microscope with similar outcome. Images can be added to the input window of the CellProfiler pipeline. After optimization, the program can be run to quantify fluorescent signal. A sample pipeline has been added as Appendix A.

Using the optimized pipeline, a 2-fold titration of Ebola virus (EBOV) was performed, and foci and cell nuclei counted. Importantly, using a 2-fold dilution series of virus, assay performance gave good linearity from a dilution of 1/400 to 1/102,400, and yielded an R^2^ value of 0.93 (Figure 1). Dilutions above and below this cut off were considered to be outside of the assay’s linear limit of detection.

We next compared the FFU assay to plaque and TCID_50_ assays. Each assay was done in 3 replicates and performed on at least two separate days. Criteria for each assay type used is shown in Table 1. A single stock of virus was used, which was thawed once and aliquoted into single use vials. Assays were performed and read by two experienced technicians. Two different plaque assays were used. CVMO stained protein in cells remained attached to the plate (Figure 2). The plaques appear as darker rings around clearances where cells have been lost due to cytopathology. For the other, the vital dye, neutral red, is taken up into viable cells and sickened cells appear as clearances (Figure 3). In contrast, the NRAO produces distinct, large clearances in the dye-stained monolayer that correspond to a loss of exclusion of the red dye (Figure 3). These are more easily identified and counted than CVMO plaques but requires greater optimization. The TCID_50_ assay was easier to set up, by serial dilutions across a multi-well plate and then staining cells remaining after virus has had time to cause cytopathology (Figure 4). However, in cases where cell death is incomplete, judgement is required to measure an endpoint. Here we used the Reed–Muench technique, where the number of wells above and below the dilution giving close to 50% of wells lacking viable cells is used to calculate the end point and titer. The FFU assay was performed as described in the methods and timed to give foci of 2–5 cells, which occurred 2 days post infection (Figure 5). FFU assays were fixed and then stained with virus-specific antibodies. The antibodies are the hardest part of the assay to optimize and reliance on commercial suppliers can sometimes be problematic. Here we used commercial monoclonal antibodies that are in good supply and gave image signal to noise ratios >2. Using the pipeline in the Appendix A, over 100 image sets can be counted in less than 30 min. The typical plate formats, amount of virus inoculum, time taken for assay incubation and processing is summarized in Table 1. Plaque assays are best done in larger plate formats, reducing potential throughput while the TCID_50_ and FFU assays can be performed in 96 to 384-well formats. 

Comparison of the assays showed consistent performance irrespective of the virus strain used. The CVMO and NRAO assays were consistently in close agreement with titers being less than 3-fold apart. In comparison, TCID_50_ titers were consistently lower, by up to 10-fold and the FFU assay titer was 10- to 15-fold higher than the plaque assays (Figure 6). These differences could be due to the nature of titer determination for the assay. For the FFU assay, virus protein is detected and stained by immunofluorescence after 48 h (2 to 3 replication cycles detected by immunofluorescence staining) while the other assays rely on cell death after 7–10 days. The difference is likely due to incubation times of virus on cells for plaque vs FFU assays with the latter being left 36–48 h and plaque assays being for 1 h. Additionally, the FFU assay measures protein expression and does not necessarily indicate cell death. FFU foci consist of small clusters of 2–5 cells expressing viral protein, as opposed to the larger number of dead cells clustered within a single plaque. 

Although assays varied in outcome based on the approach used, each showed significant differences in the magnitude of assay-to-assay variation for the two or more assays performed for each. To evaluate the variation inherent to each assay type, we generated Coefficients of Variance (CV) between counts by operator. CV was calculated in R by dividing the standard deviation between counts by the mean, and converted into percentages. Log-fold differences ranged from 0.023 to 2.371 between assay type and filoviruses assayed, demonstrating variability between titers likely due to the method in which each assay type generates a titer (Table 2). For each virus, operators counted their own assays, with variance attributed to multiple operators for a single virus. In addition, for MARV, we performed cross-counting, in which operators counted their own as well as each other’s assays as a representative of variation introduced by analysts. For NRAO plaque assay, mean percent CV was 10.99%, 41.53%, and 25.72% for EBOV, SUDV, and MARV, respectively. In CVMO plaque assay, mean percent CV was 53.67%, 37.38%, and 10.47% for EBOV, SUDV, and MARV, respectively (Figure 7). For both plaque assays, the mean of all CVs generated in each virus remained below 42%, except for CVMO in EBOV, which showed high variability with an average of 54% (Figure 7, panel 1). On average, TCID_50_ produced the highest variability, with a mean CV remaining above 30% for all viruses tested, peaking at 69% mean CV for SUDV (Figure 7). Indeed, TCID_50_ produces the highest CV seen in each virus, except for EBOV, where the CVMO plaque assay produced a higher CV (Figure 7, panel 1). In contrast to the variability seen in the plaque assays and TCID_50_, the FFU assay produced a consistently lower CV, with an average CV of 5% in EBOV and SUDV, and 9% in MARV (Figure 7, Table 1). This smaller variance is likely due to the lack of operator bias when performing counts, as well as the difficulty in making a consistent identification of Filovirus plaques versus fluorescent-based images typically being high signal:noise. To determine statistically significant differences between titers generated from different assay types, we performed a one-way ANOVA with a Turkey’s Multiple Comparison test between titers generated from each assay from a single aliquot of virus. No significant difference (*p* < 0.05) was seen between plaques assays or between plaque assays and TCID_50_. However, a significant difference (*p* < 0.05) is seen between FFU and all other assay types.

To better understand how variance in titer was defined by the counting process, we compared counts from two separate analysts, counting the same plates in assays performed by two separate operators. Since these variables were likely similar between each virus/assay type and to limit the number of assays, MARV only was used for the analysis (Table 3). Between the two plaque assay techniques, NRAO showed the highest variance from 7% to 40% variability, while crystal violet ranged from 3% to 18%. The TCID_50_ assay demonstrated the highest amount of variability, from 19% to 61% CV between titers generated by the same technician. Consistent with operator subjectivity being a larger contributor to variance, the FFU assay displayed the lowest variability (4–15%); however, the CVMO assay gave a similar range of variation 3–18% suggesting that the latter assay produces the least ambiguity of the cytopathology-based assays. As a single computer-based counter is used for FFU assays (CellProfiler), the CV shown (Table 3) is from three replicate titers for each operator for two separate operators, using the same CellProfiler pipeline.

Overall, while each assay on average gives similar outcomes, the TCID_50_ and NRAO assays have the greatest variance due to operator subjectivity while CVMO and FFU assays have lower variance.

## 4. Discussion

The steady increase in filovirus outbreaks has emphasized the need to accelerate identification and development of new therapies and candidates. Current pharmaceutical development strategies rely heavily on high-throughput screening where thousands to millions of small molecules, antibodies, and biologicals are screened for efficacy in vitro. This is followed by testing in animal models. Both in vitro and in vivo testing use assays that require minimal input of assay material, rapid yet strong statistically powered outcomes. Inside the BSL4 laboratory, techniques that reduce hands-on time inside the lab, use less virus, have a smaller footprint, and have a higher throughput are immensely beneficial. Use of mechanized and automated methods further reduces human error and variability. Reproducibility, reliability, and generation of data can be enhanced. These factors together can enable fast paced and high-throughput research in BSL-4 settings. 

To produce an assay meeting these criteria, we have developed an image analysis-based approach for counting virus infection foci. For this, we used the platform agnostic open-source software, CellProfiler [3,4]. This software is a highly flexible platform for analysis of images, which is mainly used for analysis of cell staining patterns in both 2D and 3D image sets. Our approach uses one of the simplest approaches to determining infection efficiency. Here, total and infected cells are counted as two distinct events, separately as fluorescent cell nuclei and infected cell foci in their respective fluorescence channels. More elaborate pipelines can be constructed that link cells and virus antigen levels directly and are useful if calculating the number of cells per foci or if the amount of a host protein impacts the outcome of infection [9,10]. However, for determination of virus titer, the simpler approach of counting each gives similar accuracy (not shown) but with increases in speed. In our work, using an 8 core PC, 100 image pairs can be processed in 30 min. Importantly, this occurs without significant human intervention and the pipeline algorithm is readily shared between laboratories. We find that while the FFU assay counts 10-fold more foci that are seen by plaque assays and appears 100-fold more sensitive than a TCID_50_ assay, it gives consistent outcomes with the lowest CV for the assays evaluated and does correlated with the other assay types. The simplest explanation for the difference in sensitivity between plaque and FFU assays is likely the result of leaving virus on cells for up to 48 h, instead of removing it after 1 h as is typically done in plaque assays. The difference in what is counted between each assay (small foci of cells expressing protein vs plaques of clustered cell death) can also be a cause of this difference. The reduction in TCID_50_ assay compared to the other assays may reflect the process of bulk cell interactions with viruses in wells and may be due to cell culture medium becoming depleted of nutrients or release of cell components toxic to neighboring cells but remains unclear. 

The FFU assay affords a great deal of speed, precision, and versatility. However, the need for potentially expensive antibodies and their validation can make the assay technically difficult to establish. Additionally, one must consider the mechanism by which the virus titer is evaluated. The FFU assay uses virus antigen expression, which is different to the gross cytopathology required for plaque and TCID_50_ assays. Although the two are related, filoviruses can replicate in cells for a long time, expressing virus antigen, before cytopathology is eventually seen. Cell type used or growth conditions can also impact assay outcome. For some cell types, virus antigen levels and appearance vary sufficiently to cause inconsistency in assay outcomes. Similarly, choice of Vero cells, typically low passage Vero E6 (<50 passages), is important for plaque assay outcome controlling morphology of plaques. Ultimately, it is important to evaluate what is needed from the assay and choose one that is best suited for the work. We expect the findings from this work will aid in this decision. 

Plaque assays are the benchmark for quantification of virus titers in high containment research and are a standard to compare other technologies. The relative ease of performing plaque assays, using nonspecific reagents and a direct determination of infectious virus in a sample, makes them the preferred assay for determining viral loads. However, the large format of the assay makes it cumbersome and the subjective reading of small plaques, as seen with EBOV, limits throughput and site-to-site consistency. Additionally, assays using cell death to determine viral titer require a much longer timeframe before readout than immunostaining. Immunofluorescence-based techniques can measure infectious virus loads by directly measuring viral protein, but are often overlooked due to technical challenges in finding good antibodies and in counting the small foci produced. The TCID_50_ assay allows for higher throughput than plaque assays, and generally requires less time in containment to complete as the inoculum is not removed until staining. Additionally, reagents such as crystal violet are much cheaper in a high-throughput setting. It does, however, afford the highest variability between operators and analysts (Figure 7, Table 2).

Previous studies comparing titers generated by plaque and TCID_50_ assays have shown similar variability within each assay, of about a ±5-fold differences between replicates [11,12]. Although we find a similar variation of 5–10-fold, we find that TCID_50_ titers are typically lower than plaque assay titers which was the opposite of what was previously reported for EBOV [11]. We suspect that this difference may be due to use of a recombinant EBOV that expressed GFP as an indicator of infection giving differences in cytopathology, while we have used wild type viruses but cannot rule out differences in assay set up. The same study also saw increased variability in TCID_50_ assay titers with EBOV with up to 20% higher CV when compared to NRAO which is similar to that seen in the present study. Compared to other studies that have evaluated FFU assays for filoviruses, we find a similar CV to that is reported here, remaining below 25% for both EBOV and MARV [13]. Here, we extend these findings showing similar outcomes for FFU assays for these viruses in addition to SUDV but also provide a direct comparison for all common assays and prominent filoviruses. 

In this study, we have demonstrated that titers generated by different assays, generally yield a titer within a log of each other and define the sources of variance between each. Plaque assays produce titers most similar to the other, which is expected due to the similar nature in which they derive their titer. The FFU assay generally produces a titer that is at or nearly 10-fold greater than the plaque-based assays, and is likely due to the direct measurement of viral protein, giving an earlier readout that is not dependent on long term impact on cell viability and gives higher precision. Indeed, a single viral particle for each Filovirus used is enough to produce significant cell monolayer loss after a 7–10-day incubation, as seen in the TCID_50_ assay. As such, for the FFU assay, for incubation times lasting longer than 48 h an overlay should be added. We had hoped that the comparison of titers between each assay system would enable comparison of data between studies using different assay types. This seems simplest for NRAO and CVMO plaque assays as titers do not significantly differ. Comparison of TCID_50_ assays across different reports may still be difficult to do as we saw a marked difference between the difference in plaque and TCID_50_ assay outcomes in this and another report, but may be attributed to virus type used. However, here, we saw consistent differences between TCID_50_ and plaque assay outcomes across all Filovirus types used. Lastly, to our knowledge, direct comparison of FFU to plaque assays has not been previously reported. Although there are still likely to be differences between outcomes of the FFU assay between different laboratories, we expect that use of commercial antibodies and sharing of an open-sourced method of data processing using CellProfiler will aid in comparison of future studies.

## Figures and Tables

**Figure 1 microorganisms-09-00156-f001:**
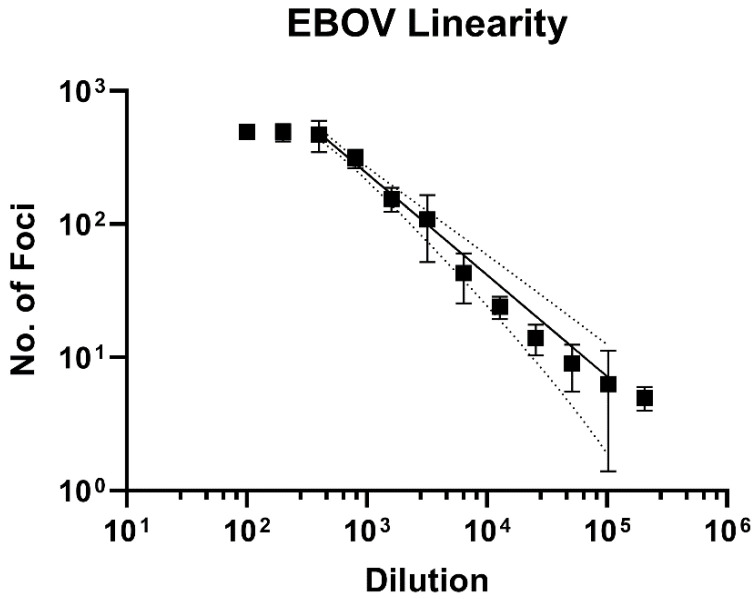
FFU assay linearity in EBOV. EBOV was used to determine FFU assay linearity. A log-log non-linear regression line was fitted to data and obtained a regression coefficient of 0.93 between 1/400- and 1/102,400-fold dilutions of the virus stock. The 95% Confidence interval is indicated by the dotted lines.

**Figure 2 microorganisms-09-00156-f002:**
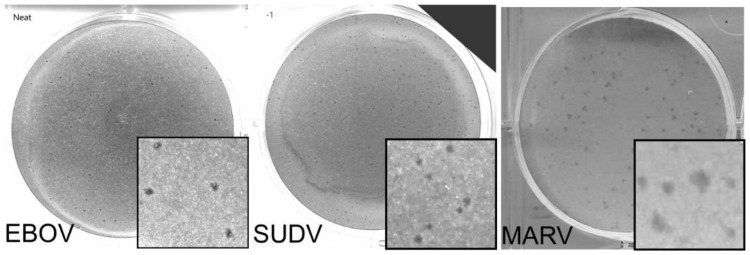
Representative images of CVMO plaque assay for EBOV, MARV, and SUDV. Here, a single well from a 6 well plate is used as a representative image for EBOV (left), SUDV (middle), and MARV (right) with insets (3.5 mm width) showing a magnified area. Images have been changed to greyscale to increase plaque visibility. Plaques may be faint, as seen in the EBOV image, or form a dark CVMO aggregate, as seen in MARV. These are counted to give the final sample titer. Dilutions used here: EBOV 1:1, SUDV 10^−1^, MARV 10^−1^.

**Figure 3 microorganisms-09-00156-f003:**
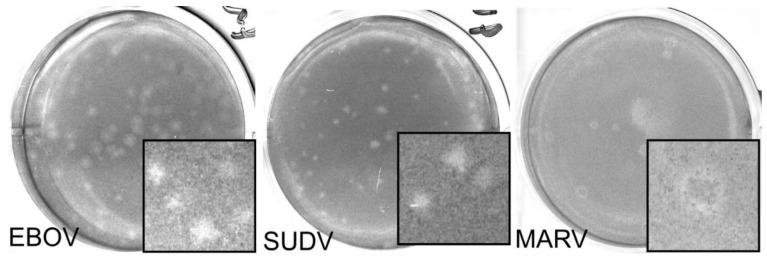
Representative images for Neutral Red plaque assays in EBOV, MARV, and SUDV. Here, a single well from a 6 well plate is used as a representative image for EBOV (left), SUDV (middle), and MARV (right) with insets (3.5 mm width) showing a magnified area. A zone of clearing can be seen in each of the representative wells pictured, although plaque size may vary. This may lead to discrepancy between counters, especially at higher titers. Dilutions pictured here: EBOV 10^−3^, SUDV 10^−3^, MARV 10^−4^.

**Figure 4 microorganisms-09-00156-f004:**
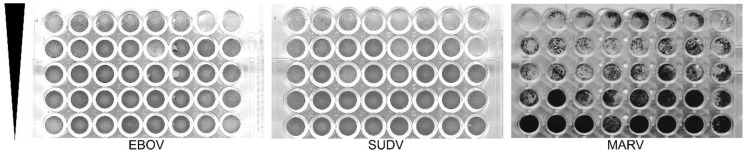
Representative images for TCID_50_ (Tissue Culture Infection Dose, 50%). Here, a representative image for EBOV (left), SUDV (middle), and MARV (right) is shown. Each column is a replicate (From the top), with each descending row increasing in dilution. The directional arrowhead denotes the virus concentration gradient, from high (top) to low (bottom) virus concentration. Titer is estimated by gross cell death within the replicates of each dilution using the Reed–Muench calculation [5].

**Figure 5 microorganisms-09-00156-f005:**
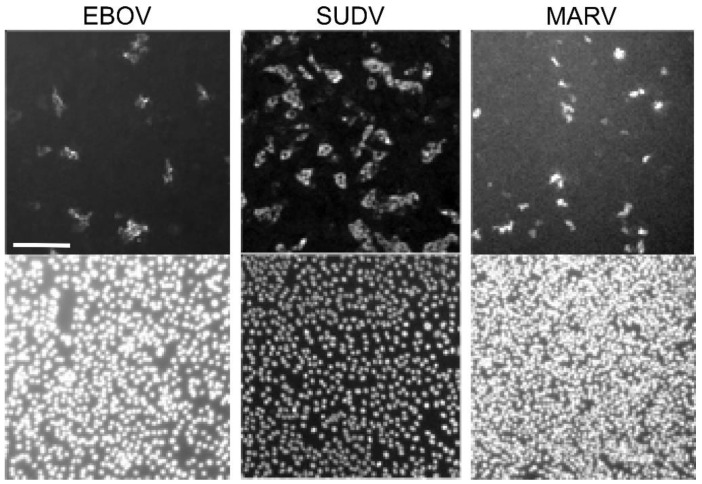
Representative images of FFU (Focus-Forming Unit) assays in EBOV, MARV, and SUDV. Here, a representative image for EBOV (left), SUDV (middle), and MARV (right) is shown. Scale bar shown on EBOV image is representative for all images, 0.2 µm. Both nuclei (Hoechst 33,342, bottom) and immunostained viral protein (top) are pictured here. Viral titers are calculated by using a free, open-source software, CellProfiler, to count foci and nuclei. Foci generally consist of 2–5 cells. Sudan virus shown here consisted of 2–8 cells per foci. For example, the full SUDV image produced a foci count of 530 and a nuclei count of 8587. The CellProfiler output image has been included as Appendix A.

**Figure 6 microorganisms-09-00156-f006:**
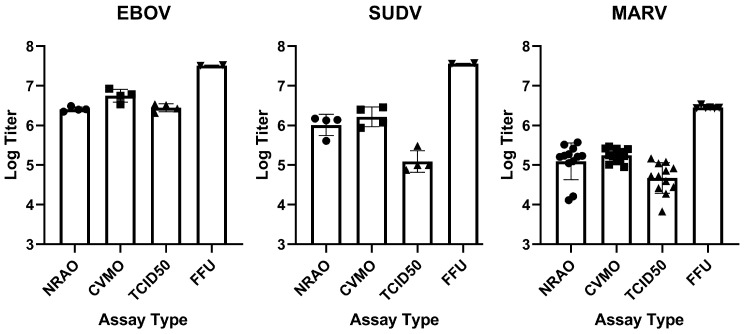
Titer comparison for the three Filoviruses in each assay. Assays were completed in EBOV (left), SUDV (middle) and MARV (right). Virus titers were determined by each assay type and compared. For NRAO and CVMO assays, titer is PFU/mL. For TCID_50_ the assay end point was determined using the Reed–Muench method to calculate when 50% of wells would show complete cytopathology. FFU is calculated from the number of infected cell foci to give FFU/mL. Mean value with standard deviation is shown. Replicates (*n*) shown: MARV *n* = 12 (NRAO, CVMO, TCID_50_), *n* = 6 (FFU). EBOV, SUDV *n* = 4 (NRAO, CVMO, TCID_50_), *n* = 2 (FFU). Symbols (•) = NRAO assay replicate, (■) = CVMO assay replicate, (▲) = TCID50 assay replicate, (▼) = FFU assay replicate.

**Figure 7 microorganisms-09-00156-f007:**
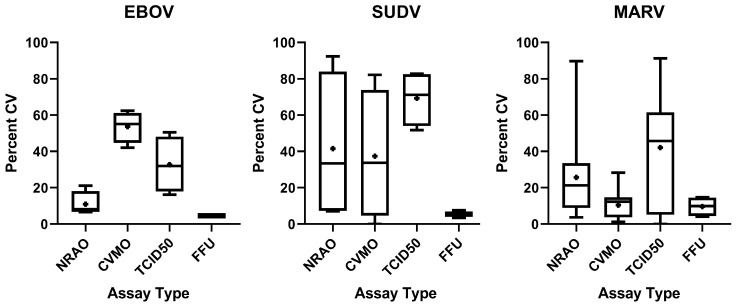
Coefficient of Variance for each assay and virus type. Assays were completed in EBOV (left), SUDV (middle), MARV (right). Assay outcomes were compared on different days and between two operators. Coefficient of Variance (CV) was determined in R Studio by dividing standard deviation by the mean and converted into a percentage. In addition, MARV CV was calculated between two operators and analysts. EBOV and SUDV CV represents variance between two operators alone. Box-and-Whisker plots display median (bar) as well as Min/Max ranges. Symbol inside each bar indicates mean.

**Table 1 microorganisms-09-00156-t001:** Assay Overview Comparison. Comparison between each viral titering method. Each assay type presents a fundamentally different method of determining viral titer, and requires differences in time and sample volume. Plate formats may vary. For the purposes of this paper, the FFU assay was done in a 384 well format. Methods for use in a 96 well format have been included in this paper.

Parameter	NRAO	CVMO	TCID_50_	FFU
Plate Format	6 Well	6 Well	96 well	384 well (Can be done in other formats including 96 well)
Volume of inoculum	100 µL	100 µL	20 µL	25 µL (384 well) 20 µL (96 well)
Challenge Incubation	1 h	1 h	10 days	2 days
Overlay/staining solution	Primary: 1:1 mixture of buffered EMEM and Agarose Secondary: 1:1 mixture of buffered EMEM containing Neutral Red (stain) and Agarose	1:5 mixture of buffered DMEM and MethylcelluloseStained with crystal violet	Stained with crystal violet	Immunofluorescent staining using virus-specific antibody
Time to complete	8 days	8–10 days	8–10 days	3 days
Read out method	Plaque count, manual	Plaque count, manual	Gross cell death within replicates, manual	Automated image analysis
Titer Calculation	Plaque count in well of lowest dilution, dilution, total volume in well	Plaque count in well of lowest dilution, dilution, total volume in well	Reed–Muench calculation	Number of infection foci stained with fluorescent antibody in lowest dilution, dilution, total volume in well
Estimated Hands-on Time, Hours (For 40 Test Points)	10	10	8	5

**Table 2 microorganisms-09-00156-t002:** Titer Comparison between Assays for 3 Filoviruses. Titer differences are shown in Log10-Fold difference. Turkey’s Multiple Comparison test was used to determine significance of difference between titers of each assay, per virus. Statistical significance was set at *p* < 0.05.

Virus	Assay Compared	Log-Fold Difference
EBOV	NRAO vs. CVMO	−0.380
	NRAO vs. TCID_50_	−0.053
	NRAO vs. FFU	−1.097
	CVMO vs. TCID_50_	0.326
	CVMO vs. FFU	−0.718
	TCID_50_ vs. FFU	−1.044
SUDV	NRAO vs. CVMO	−0.245
	NRAO vs. TCID_50_	0.873
	NRAO vs. FFU	−1.498
	CVMO vs. TCID_50_	1.118
	CVMO vs. FFU	−1.253
	TCID_50_ vs. FFU	−2.371
MARV	NRAO vs. CVMO	−0.023
	NRAO vs. TCID_50_	0.405
	NRAO vs. FFU	−1.206
	CVMO vs. TCID_50_	0.427
	CVMO vs. FFU	−1.184
	TCID_50_ vs. FFU	−1.611

**Table 3 microorganisms-09-00156-t003:** Coefficient of Variance by Counter between two assayers for MARV titers. Coefficient of Variance (CV) was calculated between counts from two different analysts, counting assays from two technicians. CV was expressed as a percentage of the standard deviation divided by the mean.

Assay	Test 1	Test 2	Test 3
NRAO			
Technician A	21.22	22.75	34.35
Technician B	40.18	6.84	10.26
CVMO			
Technician A	12.67	10.24	2.55
Technician B	8.64	10.66	18.13
TCID_50_			
Technician A	47.53	39.62	28.70
Technician B	34.50	41.74	60.86
FFU			
Technician A	14.45	4.08	7.17
Technician B	4.59	14.78	12.58

## Data Availability

Data has been provided in Appendix A and other data are available upon request to R.A.D.

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
