# Peer review of "Automation of Infectious Focus Assay for Determination of Filovirus Titers and Direct Comparison to Plaque and TCID50 Assays"

_microorganisms, 2021, doi:10.3390/microorganisms9010156_

Round 1

Reviewer 1 Report

The manuscript by Keiser and colleagues compare infectious filovirus stocks (Ebola virus, Sudan virus, and Marburg virus) using assays (plaque assays, TCID50 assays, and fluorescent focus) and different operators. With the fluorescent focus unit (FFU) assay, they have used an open-source CellProfiler platform to automate screening of samples. The authors find that the assays can have up to 10-fold differences. Overall, the manuscript is generally well written, however, the description of some of the methods require additional details such that other investigators in the field can reproduce the findings.

Major comments:

1. While the authors describe the assays in much detail, they have neglected to cite (names and manufacturer) the antibodies used in the fluorescent focus assay. Assuming other investigators will attempt to repeat the described FFU assay, it will be helpful to have the name/manufacturer of the antibodies.

2. The authors do not provide details of how the CellProfiler platform software is used in quantifying virus.

3. The authors state FFU assay is shorter in duration (2 days) compared with the other assays (8-10 days). The authors state that a caveat to the FFU assay is that it measures viral protein expression but not cell death due to virus replication. Thus a defective virus that is capable of entering the cell could express certain viral proteins but not produce infectious virus. One experiment that authors could use to address this concern would be to set up two sets of dilutions. With the first set of dilutions, perform the FFU assay to determine the FFU titer. With the second, the highest dilution that was positive by the FFU assay could be cultured for up to 10 days. If infectious virus is produced it should spread throughout the culture to cause widespread CPE. This would confirm if the more sensitive FFU assay is due to infectious virus.
Minor comments:

1. TCID50 is generally written as TCID50.

2. CO2 is generally written as CO2.

3. Line 94-95: In the sentence, “Samples were mixed well using a pipette.....,” it is unclear if pipettes are changed. Suggest changing to: “Samples were mixed well using a new pipette....”

4. Line 123: “Crystal violet” should be changed to “crystal violet.”

5. Line 130: “The wash PBS...” should be changed to “The PBS wash...”

6. Line 130: “DMEM-2/Antibiotics” should be changed to “DMEM-2/ antibiotics.

7. Line 151: “Coefficients of Variance” should be corrected to “Coefficients of variance.”

8. Line 154: “One-way” should be “one-way.”

9. Line 190: Suggest removing the word “down.”

10. Line 254: Crystal Violet should not be capitalized. Also, I disagree with the term crystal violet assay. Crystal violet is a dye that is used to stain an assay. I suggest modifying this term.

11. Line 285: Again Crystal should not be capitalized.

12. Line 315: The sentence beginning with “TCID50" should be changed to “The TCID50 assay.”

13. Line 369: The authors discuss using “low passage Vero E6." As this term could differ amongst investigators, the authors should state what they consider to be “low passage.”

Author Response

Reviewer 1 Comments

  1. While the authors describe the assays in much detail, they have neglected to cite (names and manufacturer) the antibodies used in the fluorescent focus assay. Assuming other investigators will attempt to repeat the described FFU assay, it will be helpful to have the name/manufacturer of the antibodies.

Lines 149-151: We have included both the manufacturer and the catalog number for the three primary antibodies.

  1. The authors do not provide details of how the CellProfiler platform software is used in quantifying virus.

Line 191-194: We have included a description of how images are analyzed by CellProfiler.

  1. The authors state FFU assay is shorter in duration (2 days) compared with the other assays (8-10 days). The authors state that a caveat to the FFU assay is that it measures viral protein expression but not cell death due to virus replication. Thus a defective virus that is capable of entering the cell could express certain viral proteins but not produce infectious virus. One experiment that authors could use to address this concern would be to set up two sets of dilutions. With the first set of dilutions, perform the FFU assay to determine the FFU titer. With the second, the highest dilution that was positive by the FFU assay could be cultured for up to 10 days. If infectious virus is produced it should spread throughout the culture to cause widespread CPE. This would confirm if the more sensitive FFU assay is due to infectious virus.

Line 417-419: We have added a comment addressing this. We have done previous experiments similar to what has been outlined by the reviewer, and find that even a single particle over 10 days is enough to inflict significant damage to the cell monolayer, as seen in the TCID50 and makes quantification through foci counting impossible.

Minor comments:

  1. TCID50 is generally written as TCID50.

We have changed all instanced of TCID50 to TCID50.

  1. CO2 is generally written as CO2.

We have changed all instances of CO2 to CO2.

  1. Line 94-95: In the sentence, “Samples were mixed well using a pipette.....,” it is unclear if pipettes are changed. Suggest changing to: “Samples were mixed well using a newpipette....”

We have changed the line to match the review’s recommendation.

  1. Line 123: “Crystal violet” should be changed to “crystal violet.”

All instanced of Crystal violet have been changed to crystal violet.

  1. Line 130: “The wash PBS...” should be changed to “The PBS wash...”

We have changed the line to match the reviewer’s comment.

  1. Line 130: “DMEM-2/Antibiotics” should be changed to “DMEM-2/ antibiotics.

We have changed this line to match the reviewer’s comments

  1. Line 151: “Coefficients of Variance” should be corrected to “Coefficients of variance.”

We have changed this line to match the reviewer’s comments

  1. Line 154: “One-way” should be “one-way.”

We have changed this line to match the reviewer’s comments

  1. Line 190: Suggest removing the word “down.”

We have changed this line to match the reviewer’s comments

  1. Line 254: Crystal Violetshould not be capitalized. Also, I disagree with the term crystal violet assay. Crystal violet is a dye that is used to stain an assay. I suggest modifying this term.

We have changed the name of the crystal violet assay to the Crystal Violet Methylcellulose Assay (CVMO) to address this issue, and have changed all instanced of “crystal violet assay” to “CVMO assay”

  1. Line 285: Again Crystal should not be capitalized.

We have changed this line to match the reviewer’s comments

  1. Line 315: The sentence beginning with “TCID50" should be changed to “The TCID50 assay.”

We have changed this line to match the reviewer’s comments

  1. Line 369: The authors discuss using “low passage Vero E6." As this term could differ amongst investigators, the authors should state what they consider to be “low passage.”

We have added the suggested maximum passage number considered to be at “low passage”

Reviewer 2 Report

Keiser et al. performed a detailed comparison between three assays used to measure infectious virus for filoviruses. Manuscript is well written, data clearly presented and provides useful data for establishing these types of assays at other institutions. Some key details are needed before publication. 

Comments:

Line 368: Authors state that low-passage Veros are essential for plaque morphology, authors should state in methods what passage of Veros they used.

Line 146: Sources of these antibodies need to be provided, catalog numbers would be ideal. 

How were the images for the FFU assay acquired? What microscope? 

Figure 5: Can authors provide an example of what CellProfiler counts on those images? How many foci are in these images? An important detail missing is what do the authors or software settings define as a cluster of cells representing one foci versus two close together foci? For example in the top right of the SUDV image, there is a "chain" of antigen positive cells, is this defined as one foci or several foci in a line? This can be a source of "subjectivity" in these automated cell counting setups. 

What is the replication time of these filoviruses? Authors did not appear to use an overlay on the FFU assay, thus could the 10-fold increase in titer of the FFU assay be due to production and release of infectious virus into the media, infection of new cells and subsequent antigen production? Line 259 - 263: Authors speculate that the difference in titers is "likely" due to length of time inoculum is left on the cells but do not provide any supporting evidence. This could be tested by not removing inoculum during the plaque assay and adjusting the overlay to account for this volume. "Likely" should be replaced with "could be". It is also unclear why foci of 2 - 5 cells versus clusters of dead cells in plaques would alter the calculated titer. 

Line 257: Authors state that the virus protein is detected after 48 hours. This is somewhat misleading as authors fix the plates after an overnight incubation. Thus although they technically read the plates after 48 hours, they're detecting antigen produced after just an overnight incubation not 48 hours of viral replication. 

Table 1 could be improved by addition of something like an "Estimated Hands on Time" for an arbitrary amount of plates. 

Author Response

Reviewer 2 Comments:

Line 368: Authors state that low-passage Veros are essential for plaque morphology, authors should state in methods what passage of Veros they used.

We have added the suggested maximum passage number considered to be a “low passage”.

Line 146: Sources of these antibodies need to be provided, catalog numbers would be ideal. 

Lines 149-151: We have included both the manufacturer and the catalog number for the three primary antibodies.

How were the images for the FFU assay acquired? What microscope? 

Line 191-192: We have added a statement of how images were acquired and the types of microscope used for this manuscript

Figure 5: Can authors provide an example of what CellProfiler counts on those images? How many foci are in these images? An important detail missing is what do the authors or software settings define as a cluster of cells representing one foci versus two close together foci? For example in the top right of the SUDV image, there is a "chain" of antigen positive cells, is this defined as one foci or several foci in a line? This can be a source of "subjectivity" in these automated cell counting setups. 

We have added an example count image as a supplemental figure (S2) of the CellProfiler ouput, as well as the count for the supplemental figure in the description of figure 5 (line 260-261).

What is the replication time of these filoviruses? Authors did not appear to use an overlay on the FFU assay, thus could the 10-fold increase in titer of the FFU assay be due to production and release of infectious virus into the media, infection of new cells and subsequent antigen production? Line 259 - 263: Authors speculate that the difference in titers is "likely" due to length of time inoculum is left on the cells but do not provide any supporting evidence. This could be tested by not removing inoculum during the plaque assay and adjusting the overlay to account for this volume. "Likely" should be replaced with "could be". It is also unclear why foci of 2 - 5 cells versus clusters of dead cells in plaques would alter the calculated titer. 

Line 268-269: We have added the replication time of the virus seen by fluorescent staining. Line 419-420: We have addressed the reviewer’s comment regarding inoculum removal, as any FFU assay with an incubation time greater than 48 hours should include an overlay. We have also changed “likely” to “could be” as requested. The difference in cell titer is likely due to the amount of time inoculum remains on the cells, in addition to the method of counting (Line 368-371)

Line 257: Authors state that the virus protein is detected after 48 hours. This is somewhat misleading as authors fix the plates after an overnight incubation. Thus although they technically read the plates after 48 hours, they're detecting antigen produced after just an overnight incubation not 48 hours of viral replication. 

The incubation time for assay should have read 36-48 hours, and we have adjusted all instances of the incubation time for FFU to reflect this.

Table 1 could be improved by addition of something like an "Estimated Hands on Time" for an arbitrary amount of plates. 

Added “Estimate Hands-on Time” to Table 1.

Reviewer 3 Report

The authors compared the various methods available to titer live filoviruses and carefully compared their variances. In addition they created a valuable automated FFU counting protocol for the filovirus community.

Author Response

Reviewer 3 Comments:

No Comments listed.